# Functional Status of Patients over 65 Years Old Intervened on for a Hip Fracture One Year after the Operation

**DOI:** 10.3390/healthcare11101520

**Published:** 2023-05-22

**Authors:** Pablo A. Marrero-Morales, Enrique González-Dávila, María Fernanda Hernández-Gutiérrez, Eva M. Gallego-González, Martina Jiménez-Hernández, Emilio J. Sanz-Álvarez, Natalia Rodríguez-Novo, Yurena M. Rodríguez-Novo

**Affiliations:** 1Department of Physical Medicine and Pharmacology, Faculty of Health Sciences, University of La Laguna, 38320 Santa Cruz de Tenerife, Spain; 2Department of Matemáticas, Estadística e Investigación Operativa, Instituto IMAULL, University of La Laguna, 38320 La Laguna, Spain; egonzale@ull.edu.es; 3Traumatology Service, Complejo Hospitalario Universitario de Canarias, 38320 La Laguna, Spain; dra_egallego@yahoo.es; 4Medicine Section, University of La Laguna, 38320 La Laguna, Spain; martinajimhdez@gmail.com; 5Clinical Pharmacology Service, Complejo Hospitalario Universitario de Canarias, University of La Laguna, 38320 La Laguna, Spain; esanz@ull.edu.es; 6Nursing Section, Faculty of Health Sciences, University of La Laguna, 38200 Santa Cruz de Tenerife, Spain; nrodrigu@ull.edu.es (N.R.-N.); yrodrign@ull.edu.es (Y.M.R.-N.)

**Keywords:** sarcopenia, hip fracture, functional physical performance, functional status, geriatrics

## Abstract

Objectives: Evaluation of the functional status one year after a hip fracture surgery and the influence of sarcopenia and other clinical factors at the time of admission. Method: Prospective observational study with 135 patients over 65 years of age. Functional status of basic (modified Katz) and instrumental activities (Lawton and Brody) and walking ability (Functional Ambulation Classification, FAC) was measured on admission, at discharge, and telephonically one year later. The risk or positive screening of sarcopenia (SARC-F) and cognitive status (Pfeiffer), as well as clinical variables, were evaluated. Results: 72% of patients are women; 36% have a risk of sarcopenia (Sarc-F ≥ 4), and 43% have moderate–severe cognitive impairment (Pfeiffer ≥ 5). Walking capacity at one year was closer to the values at admission more often in women than in men (0.2 ± 1.3 points vs. 0.9 ± 1.6; *p* = 0.001), as well as in patients without risk of sarcopenia versus sarcopenic patients (0.3 ± 1.2 points vs. 0.7 ± 1.7; *p* = 0.001), although their evolution did not show significant differences (*p* = 0.183). Instrumental activities after one year have not been recovered (1.7 ± 2.5 points; *p* = 0.032), and patients at risk of sarcopenia showed worse values (1.7 ± 1.9 points vs. 3.7 ± 2.7; *p* < 0.001) and worse evolution (*p* = 0.012). The evolution of basic activities varied according to the risk of sarcopenia (0.6 ± 1.4 points vs. 1.4 ± 2.1; *p* = 0.008). Conclusions: Functional status at one year is related to the functional status at admission, the positive screening of sarcopenia, sex, and cognitive impairment of the patient. Knowing at the time of admission an estimate of the functional status at one year will help to reinforce the individual treatment of patients with a worse prognosis.

## 1. Introduction

Hip fracture in the elderly is one of the main causes of hospital admissions. It is estimated that by 2025 globally, the figure will be 2.5 million, with a greater increase since 1990 in the case of men (310%) compared to women (240%). The main demographic changes will be established in Asia, where the population of adults over 80 years of age in 2025 will be 50%, with forecasts of 37% of all fractures occurring in the world [1].

Hip fracture has a negative impact on functional recovery after one year of evolution, aggravated in situations of decreased cognitive status, advanced age, delirium, and the presence of comorbidities. These situations are usually associated with less mobility, loss of activities and quality of daily life, a decrease in appendicular muscle mass, and an increase in body mass index [2,3].

Sarcopenia [4] is a geriatric syndrome that affects the musculoskeletal system, characterized mainly by a decrease in strength, mass, and muscle quality, which reduces the patient’s physical performance [5]. It can generate a mobility disorder [6] of the basic and instrumental activities in daily life, as well as a general decrease in life quality [7], increasing morbidity and mortality [8] of those affected. Rosenberg [9], in 1989, was the first to refer to this term as a relationship between an individual’s functional status, body composition, independence, and its adverse consequences, especially regarding elderly individuals. The European Working Group on Sarcopenia in elderly patients (EWGSOP2) [5] defined it in 2018 as a muscle disease (muscle insufficiency) caused by adverse changes in the muscle accumulated [10,11].

The tests and cut-off points of sarcopenia were established by EWGSOP2 via a sequential algorithm. In the first phase of this algorithm, the Sarc-F [12] test is used as an initial screening for clinical suspicion of sarcopenia risk. If the Sarc-F is positive, other tests will be evaluated and performed [13,14] to confirm the final diagnosis. Different physical performance tests [15] can be used to assess the severity of sarcopenia.

The presence of sarcopenia increases the risk of hospitalization and cost of care [16,17]. Patients with sarcopenia are associated with worse outcomes after undergoing hip surgery. This makes early detection and diagnosis of the disease essential to avoid the significant personal, social, and economic burden of not treating it [18]. The evidence indicates that the risk factors for sarcopenia can be modifiable and can be prevented with early and effective interventions [10].

With age, the basic and instrumental functional activities of people, as well as the ability to walk, tend to decrease [19]. The main objective of this study is to evaluate the functional status one year after hip fracture surgery, relating it to the risk of sarcopenia, cognitive impairment, baseline functional status, and other possible factors associated with recovery (sex, age, body mass, delirium, and comorbidities) at the time of admission.

## 2. Materials and Methods

A prospective observational study was developed at the Complejo Hospitalario Universitario de Canarias (CHUC), including all patients who underwent hip fracture surgery over 65 years of age and who were admitted to the Traumatology Surgery Service from September 2019 to December 2021. The sample size, *n* = 135 patients, was obtained by establishing that the functional recovery and return to the starting values in the functional evaluation tests was less than 0.25 points with a standard deviation of the difference (before—after) of 1.0 points, with the standard confidence and power levels of 95% and 80%, respectively, and adding 5% for possible loss to follow-up.

Those patients who did not meet any of the following criteria were excluded: non-mobile prior to admission, not authorized for weight-bearing of lower extremities, patients with hip fracture due to tumors, those with upper extremity fractures, patients with stroke or sequelae of this pathology, patients with severe cognitive impairment and who were not able to collaborate, or those who failed to sign the informed consent for participation (Appendix A).

### 2.1. Procedure

The study collected information over three periods: at admission, at hospital discharge, and telephonically, at home, or residence one year after the intervention. All data were collected by the same research physiotherapist (first author).

### 2.2. Assessment at the Admission

On the first day of physiotherapy treatment, the patient, or the primary caregiver, was requested with signed consent. Once the patient’s participation was accepted, sociodemographic information and baseline functional status (two weeks before the fracture) on basic and instrumental activities of daily living and walking ability to be developed at the place of residence were collected. In those patients with moderate or severe cognitive impairment, the questionnaires were usually filled out in the presence of a responsible family member or the main caregiver. Additionally, cognitive function and the risk of sarcopenia were assessed.

The basic activities (bathing, dressing, toileting, transferring, continence, and feeding) were studied according to the modified Katz Index [20] with a score between 0 and 6 points. A value between 0 and 2 is categorized as severe disability, between 3 and 4 as moderate disability, and between 5 and 6 as mild or no disability.

Instrumental activities of daily living (ability to use telephone, shopping, food preparation, housekeeping, laundry, mode of transportation, responsibility for own medications, and ability to handle finances) were assessed using the Lawton and Brody Scale [21] (Lawton) with a score between 0 (severe dependency) and 8 (independent) points. For women, it consists of eight total items. For men, the items addressed in food preparation, housekeeping, and laundry are not counted, with the maximum score being 5 points. For statistical comparison purposes, the data was fitted to the same scale with a maximum score of 8 points. From 0 to 2, the patient presents a severe dependency; between 3 and 5, a slight dependency, and between 6 and 8, the patient is independent.

Walking ability (non-functional ambulator, dependent on physical assistance continuously, dependent on physical assistance intermittently, dependent on supervision, independent level surface only, and totally independent) was assessed using the Functional Ambulation Classification [22] (FAC) with a score between 0 (great physical help) and 5 (walking independently) points. Between 0 and 1, the patient walks with aid; between 2 and 3 points with supervision, and between 4 and 5 points, the patient walks independently.

Quantitative cognitive function was evaluated according to the Pfeiffer [23] questionnaire. The score varies between 0 (intact intellectual functions) and 10 (severe intellectual impairment) errors, classifying it as intact intellectual functions when it presents between 0 and 2 errors, mild intellectual impairment between 3 and 4 errors, moderate intellectual impairment between 5 and 7 errors, and severe intellectual impairment between 8 and 10 errors.

The risk of sarcopenia (strength, assistance in walking, rising from a chair, climbing stairs, and number of falls in the past year) in the sample was measured with the Sarc-F [24,25] test. The risk of sarcopenia is considered when the total score is ≥4.

The presence of delirium is diagnosed by a geriatrician according to the criteria of the Confusion Assessment Method (CAM) [26]. The domains measured by this method are acute change in mental status, inattention, disorganized thinking, altered level of consciousness, disorientation, memory impairment, perceptual disturbances, psychomotor retardation, and altered sleep–wake cycle.

After surgery, all patients were subjected to daily exercises by the physiotherapist–researcher. Specific exercises were added depending on the characteristics of each patient and the different types of osteosynthesis materials used in hip fractures.

### 2.3. Discharge Evaluation

On the day of hospital discharge, functional status was reassessed using the aforementioned tests. The total distance covered by the patient with the walker out of a total of four free walks was also assessed. The beginning and the end of the walk were performed in a bipedal position with feet at the same distance. This variable will be identified as “discharge walking distance”. All patients who were already in an elderly residence returned to their center. Of the patients who were at home, all but three returned home to be cared for by their families. These three, who did not return home, were admitted to an elderly residence.

### 2.4. Status at One Year

Functional status was assessed one year after hospital discharge by means of a follow-up telephone interview, where the ability to walk and basic and instrumental activities of daily living were measured according to the aforementioned scales. When the patients were not able to answer the tests, the response of their family members or caregivers was accepted. Patients were assessed exclusively to collect functional data (other information during that period is lacking). All patients follow the established rehabilitation protocol in their primary care centers.

### 2.5. Statistical Analysis

The information is presented showing the mean ± standard deviation, median and interquartile range, and absolute frequencies and percentages, as required.

The comparison of the results about sex, risk of sarcopenia, or death was carried out using the contrasts of the *t*-student, U-Mann–Whitney, Chi-square, or Fisher’s exact test. The relationship between continuous variables was carried out with the Pearson correlation coefficient.

Multiple linear regression was applied to study functional test scores one year after discharge based on the predictor variables of demographic information (sex and age), factors associated with recovery (risk of sarcopenia, categorized Pfeiffer score, delirium, type of fracture, and Charlson index) and functional test scores collected at admission. Variable selection was performed using the stepwise method with *p*-in = 0.05 and *p*-out = 0.10.

The evolution of functional tests collected over the three periods of time (admission, discharge, and one year), according to the predictor variables sex and risk of sarcopenia, was carried out with repeated measure analysis (functional test given in points as outcome) and Generalized Linear model with multinomial distribution (categorized functional tests as outcome). The linear effect of time shows the difference between the evaluation at admission and after one year, while the quadratic effect shows the difference between the evaluation at discharge and the mean of admission after one year. The interactions time by sex or by the risk of sarcopenia evaluate whether the evolution over time between groups differs from one another.

All analyses were carried out with SPSS v.25 (IBM SPSS Statistics) and MedCalc v19.5. The results were considered significant when the *p*-value was less than 0.05, corrected by the Bonferroni method.

## 3. Results

### 3.1. Information at Admission

The general characteristics of the total population evaluated at admission and discharge are shown in Appendix A. Initially, it was composed of 135 patients; 97 (71.9%) were women, and 38 (28.1%) were men. The mean age of women was 82.2 ± 7.4 years and that of men 80.2 ± 8.0 years, showing no significant differences between them. A total of 45% of the patients presented pertrochanteric fracture and 42% intracapsular fracture. The most used osteosynthesis material was the intramedullary nail (61%), followed by partial hip replacement (34%).

The functional tests, the sarcopenia risk test, and the cognitive impairment questionnaire show a general trend toward a worse state of women than men. Moderate or severe disability to carry out basic daily tasks occurs in 33% of women compared to 16% of men (*p* = 0.015). Dependence to carry out instrumental activities occurs in 59% of women compared to 48% of men, with 50% scoring higher than 5 and 6.4 points, respectively (*p* = 0.008). Supervision or assistance when walking is required in 16% of women versus 5% of men (*p* = 0.086). The Sarc-F test showed a mean score of 3.3 ± 2.5 points in women compared to 2.0 ± 1.9 in men (*p* = 0.004), with 40 (41%) of the women at risk of sarcopenia versus 8 (21%) of the men (*p* = 0.030). The evaluation of the cognitive impairment did not show significant differences between sexes, with 43% of the patients being classified as having moderate or severe deterioration. However, patients at risk of sarcopenia have greater cognitive impairment than patients without risk (*p* = 0.008). In the group at risk, 56.3% presented moderate–severe cognitive impairment compared to 34.5% of the group without risk (*p* = 0.018). Additionally, in this same risk group, the functional tests evaluated show worse values; age is higher; the percentage of women is higher, and the Charlson comorbidity index also shows significantly worse values compared to the group without risk (Table 1). There is no significant difference between these groups in body mass index, type of fracture, osteosynthesis material, percentage of delirium, or place of origin. Both the number of physiotherapy sessions and the days of hospital stay are significantly higher in the group at risk.

The scores of the three functional tests are positively related to each other. Between Katz and Lawton, r = 0.436; between Katz and FAC, r = 0.540 and between Lawton and FAC, r = 0.482 (all *p* < 0.001). Regarding the relationships with the Sarc-F test, there is an inverse relationship with the functional status provided by the Katz, r = −0.505, by the Lawton, r = −0.442, and with the FAC, r = −0.569 (all *p* < 0.001).

A total of 17 (12.6%) patients died during the follow-up period. The functional test scores at admission, Sarc-F and Pfeiffer, were compared between the deceased and the 113 patients who completed the follow-up. No significant differences were found between these two groups (Katz: 4.65 ± 0.86 vs. 4.51 ± 1.11; Lawton: 4.24 ± 2.31 vs. 4.26 ± 2.54; FAC: 4.18 ± 0.53 vs. 4.25 ± 0.84, respectively). In the analysis, it was found that in 10 of the 17 deceased, 59% were at risk of sarcopenia compared to 32% who completed the follow-up (*p* = 0.030). Likewise, 11 of the deceased (65%) presented moderate or severe cognitive impairment compared to 39% of the non-deceased (*p* = 0.043).

### 3.2. Follow-up Study Based on the Risk of Sarcopenia

Of the 135 patients initially included in the sample, 113 (83.7%) completed the follow-up period. Of the 22 (16.3%) who did not complete it, at least 17 died (12 women and 5 men), and follow-up was not possible in 5 cases. As for patients with the follow-up, all answered the telephone; 19 patients answered personally, and 94 (83%) were the caregiver. Seventeen (15.1%) patients resided in geriatric or social health centers.

A total of 36 patients (31.9%) were classified with a risk of sarcopenia. The distribution by age and sex (*p* = 0.012 and *p* = 0.074, respectively) showed older age (3.4 years, CI95% 0.4, 6.5) and a higher percentage of women (83% vs. 66%) in the group with a risk of sarcopenia than in the group without risk.

The Pfeiffer test score was 1.0 (CI95% 0.04, 2.0; *p* = 0.042); points were higher in patients at risk of sarcopenia than without it. In patients without risk of sarcopenia, 34% had severe or moderate cognitive impairment compared to 53% in the group with risk (*p* = 0.262).

The discharge distance test shows statistical significance depending on whether patients were at risk of sarcopenia or not. Patients with a risk of sarcopenia walked an average of 5.9 ± 6.4 m. compared to 30.7 ± 16.9 m. of those without risk (*p* < 0.001). 83% of the former covered less than 10 m, compared to 3% of the patients without risk (*p* < 0.001). According to the Youden index, the best point that discriminates between these two groups is the distance of 12.3 m. (*p* < 0.001).

Table 2 shows the values of the different functional tests performed according to the initial sarcopenia risk classification. The linear effect of the FAC (*p* = 0.394) and the Katz (*p* = 0.077) indicate that the mean values one year after surgery do not differ significantly from those observed at admission. In the case of Lawton, the evaluation of the instrumental activity of daily life, the starting values have not yet been reached (*p* = 0.032).

The interaction study shows that the evolution of the index of basic activity in the daily life of men and women has been similar (*p* = 0.118, Figure 1a), although the evolution depending on whether they are at risk of sarcopenia does differ (*p* = 0.008, Figure 1b). In the case of Lawton, both sexes (Figure 1c) are at risk of sarcopenia (Figure 1d), and differences in evolution are observed (*p* = 0.001 and 0.012, respectively). Regarding sarcopenia, it is observed that recovery is faster in the group without risk. Regarding walking capacity, women recovered quicker than men after hospital discharge (*p* = 0.001, Figure 1e), while the evolution over time of patients with a risk of sarcopenia does not differ significantly from one observed in those who did not present said risk (*p* = 0.183, Figure 1f). Nevertheless, values were always lower in the case of those who presented a risk.

### 3.3. Functional Status at One Year

The Lawton and Brody scale scores at admission and after one year of evolution have the highest correlation, r = 0.605 (*p* < 0.001). In the case of the Katz index and FAC walking capacity, although the positive relationship remains, they present lower correlations, r = 0.272 (*p* =0.004) and r = 0.363 (*p* < 0.001), respectively.

Table 3 shows the variables that are related to the scores of the functional tests one year after the hip operation, according to the admission information. As for the Katz index (R^2^ = 0.739), it was established that the final score is affected by whether there is a risk of sarcopenia (Sarc-F ≥ 4), producing an average reduction of 1.39 ± 0.28 points (*p* < 0.001), by the score of the Pfeiffer (*p* = 0.029), and the presence of delirium. The final Lawton and Brody score (R^2^ = 0.862) showed an inverse relationship with the categorized Pfeiffer score, risk of sarcopenia, age, and delirium (*p* < 0.001). Regarding the FAC (R^2^ = 0.713), the results indicate that it influences the risk of sarcopenia, producing a mean reduction of 1.22 ± 0.26 points (*p* < 0.001); men obtain lower scores than women (0.84 ± 0.27 points, *p* = 0.003), and the presence of delirium, producing a mean reduction of 1.34 ± 0.50 points (*p* = 0.009). As internal validation of these three models, Appendix A shows the residuals (observed minus estimates) against the observed values one year after the hip operation.

As an example of application, suppose we have an 80-year-old male patient with a risk of sarcopenia and with a Pfeiffer score of 3 points (mild cognitive impairment). Applying the models in Table 3 for Katz, Lawton, and FAC, a score of 3.1, 2.5, and 2.8 points, respectively, if the patient does not have delirium, and 2.0, 0.7, and 1.5, yes they present it, would be obtained. Thus, the estimated value of the FAC would be obtained by performing the operation 4.715 − 1.224 − 1.342 + 0.1860 = 1.492, using the coefficients of the model in Table 3.

## 4. Discussion

Our study relates the risk of sarcopenia at admission to the functional status of elderly patients with hip fractures one year after the operation. The Sarc-F test is a simple and quick test that provides information regarding the risk of sarcopenia [27]. When the risk of sarcopenia is incorporated into multiple linear regression models, the functional status of the patient is better explained. The factors associated with a worse functional recovery in patients after a hip fracture, older age, worse cognitive status, presence of delirium, and a higher rate of comorbidity, results that coincide with those shown in Mayoral et al. [3], are closely related to the risk of sarcopenia (Table 1). This explains that in several of the models presented, once the risk of sarcopenia is selected to enter, most of these factors are no longer present except for delirium.

This research presents a higher percentage of women with hip fractures than men, as in previous studies [25]. Ha, Y-C et al. [25] also indicate that the Sarc-F test is an economical and convenient tool.

In our study, the baseline functional status of women shows a general tendency toward a worse functional status than men (Appendix A). This was significantly observed in the evaluation of basic (Katz) and instrumental (Lawton) activities of daily living, as well as with the risk of sarcopenia (Sarc-F). A trend close to significance was also observed in the evaluation of ambulation ability (FAC). These findings coincide with the work of Meskers et al. [28]. Wu, Tai-Yin et al. [29], in a prospective study using the Sarc-F test, also found that the participants with a positive screening of sarcopenia were mostly women.

Not all studies relate hip fractures to the positive screening of sarcopenia measured by the Sarc-F test. Several studies use dual-energy *X*-ray absorptiometry (DXA) [13] or other systems, such as bioimpedance [30], for the diagnosis of sarcopenia. In these works, the results show a special vulnerability to sarcopenia in men, unlike the results of studies that use the risk of sarcopenia. This differentiating fact may be associated with the measurement of the appendicular muscle mass of men, as indicated by Landi et al. [31] and Di Monaco et al. [32].

At the time of admission, the patients with the positive screening of sarcopenia in this study already presented lower values in the scores of the three functional tests evaluated compared to the group without risk (Table 1), coinciding with other investigations [33,34,35]. In particular, 77.5% of women versus 62.5% of men with a risk of sarcopenia were dependent on instrumental activities, and 45% and 37.5%, respectively, for basic activities. Wearing et al. [35] found that patients with probable sarcopenia were 2.8 times more likely to be dependent on basic life activities than those who did not, in line with the results found. However, Tanimoto et al. [34] found that the percentage of disability in patients with sarcopenia for instrumental activities was 39% in men compared to 30.6% in women.

The percentage of patients at admission with cognitive impairment (Pfeiffer ≥ 3) in this study is 67% (43% with moderate–severe cognitive impairment). These percentages vary depending on the bibliography consulted. Mayoral et al. [3] find that 55% present cognitive impairment, and Solsona Fernández et al. [36] think that it is 61.5%. These types of variations may be due to the medication supplied and protocols followed in different centers. We are not aware that the hip fracture itself affects cognitive function one year later, although it is possible that the incidence of pathologies could worsen or be more prevalent after the hip fracture [37]. This could lead to a change in daily activity performance and, ultimately, have an impact on cognitive function. All could be additive to the effect of aging itself.

### Status One Year after the Surgery

The patients who died during the follow-up period already had a higher percentage of risk of sarcopenia and a moderate–severe cognitive impairment than the patients who completed the follow-up. Both basic activities of daily life and instrumental activities present deficits one year after intervention in the groups defined according to the risk of sarcopenia, being more evident in the patients with sarcopenia. Only in the ambulation ability test (FAC) the analyzed patients did not present statistical significance with regard to the values prior to the fall. These results concur with those described by Lim and SK. et al. [38].

In our research, the walking ability score showed that patients with a risk of sarcopenia always presented lower values than patients without that risk (Table 1). This behavior continued without significant variations throughout the periods observed. Steihaug et al. [39], using the New Mobility Score [40], showed that sarcopenia is associated with less mobility before the fracture, 3 months, and one year after surgery, compared to the results of the non-sarcopenic group. These same results were obtained by Chen and Yu-Pin et al. [2], who carried out a one-year follow-up after hip surgery, assessing the instrumental activities of the daily life of the patients using the Barthel test in different recovery periods. In addition to a loss of daily activities at one year, sarcopenic patients decreased their skeletal muscle mass and increased their body mass index and total body fat.

Our multiple linear regression model for the Lawton and Brody scale (Table 3) shows that the degree of recovery of instrumental activities decreases with age. This coincides with the follow-up study evaluating the functional status of the Barthel index carried out by Mayoral et al. [3]. The patient’s cognitive status, according to the Pfeiffer index, and the presence of delirium help to explain in our models the functional status of basic and instrumental activities, such as in the study proposed by Mayoral et al. for the Barthel index. The patients who obtain better scores in the models (Table 3, Katz, Lawton, FAC) could shorten or avoid rehabilitation at health centers, and those with worse results could benefit from prolonged physiotherapy in specialized rehabilitation centers or at home. This information can be discussed in advance with the family, along with the expected functional prognosis, in order to anticipate the need for ortho prosthetic aids or to seek a residential facility in case of need.

From the medical point of view, care objectives can be individualized according to the functional prognosis, and further studies to confirm or assess sarcopenia could be performed based on SARC-F results. Our results could allow nurses, doctors, and physiotherapists to anticipate the functional status and needs of patients in advance, providing valuable information also for prevention and health promotion [31].

Gherardini et al. [41] predict functional status according to the Barthel index at one year, using walking speed over a short distance. Duke et al. [42] used the distance covered on the second day after surgery and the help required to get from the bed to sit down as predictors of transfers, ambulation, and level of mobility after a hip fracture. In our study, a strong relationship was also observed between the total distance walked and having or not having a positive screening for sarcopenia according to the Sarc-F. Heiden et al. [43] conclude that early ambulation after hip fracture surgery is associated with decreased postoperative mortality.

The limitation of this study is the need to validate these multiple linear regression models with external data to be able to use them as predictive models. Regarding the instrumental activities of daily life measured according to the Lawton scale, several authors mention that it is more useful in the case of women since many men do not perform the functional activities studied [44]. It is common to use the functional assessment scales in person; in this study, the measurements one year after the hip operation were made by telephone interview, which may cause some bias [45]. Furthermore, this may be increased since the data collection is carried out by the same researcher, who, therefore, knows the status of the patients. Given that the rehabilitation treatments after hospital discharge are carried out in the primary care centers that correspond to the patient, the influence of unexplored factors in our results cannot be ruled out. That is, acute events during follow-up, such as hospitalization or stroke, private rehabilitation interventions, etc. Although a nutritional structured assessment was not performed, no differences in BMI were found between both SARC-F groups. However, the interference of nutrition status on our results cannot be discarded and should be explored. A possible improvement of these models could be the incorporation of other sarcopenia measurement methods, such as DXA, grip strength, physical performance, etc. However, all these methods require specific and costly instrumentation that would make their use in routine clinical practice more difficult.

## 5. Conclusions

The proposed models make it possible to relate the functional status one year after the hip operation with information at the time of admission. From the assessment of these functional tests at the time of admission, the evaluation of the risk of sarcopenia, the patient’s cognitive status, factors associated with recovery, and demographic variables, such as age and sex, it is possible to obtain reliable information concerning evolution. The results obtained can be used in routine clinical practice with the intention of evaluating, at the time of admission, the possible evolution of the functional status of an elderly patient who has had a hip fracture or could have one. This will make it possible to reinforce the individual treatment of patients with a worse prognosis, including them in specific rehabilitation programs [31] and improving daily clinical practice and intervention models for patients.

## Figures and Tables

**Figure 1 healthcare-11-01520-f001:**
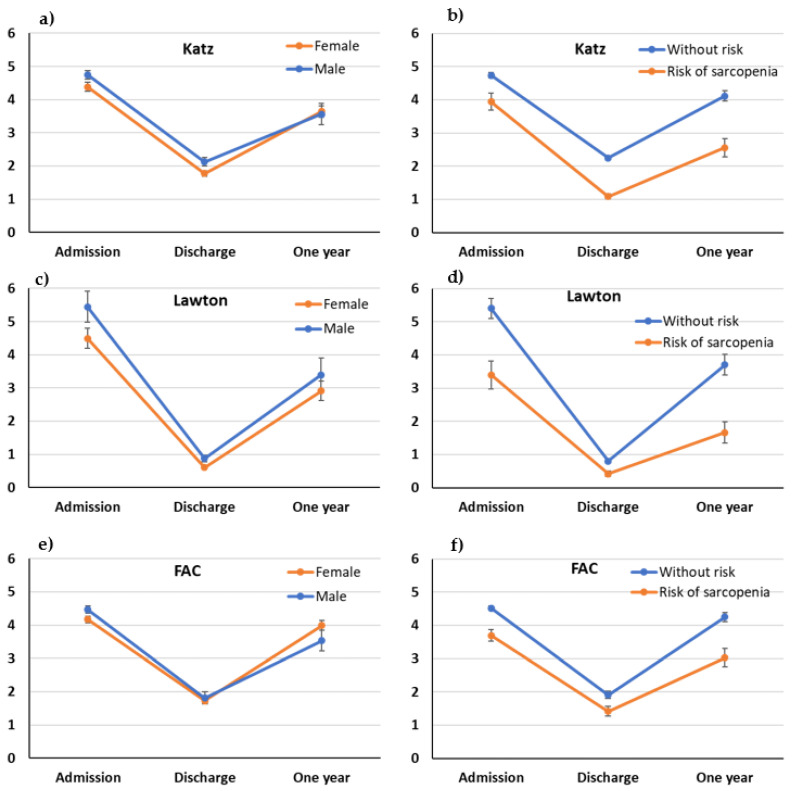
Change in functional tests (Katz: (**a**,**b**); Lawton: (**c**,**d**); and FAC: (**e**,**f**) respectively) over time according to sex and risk of sarcopenia (dots represent mean score, and bars represent a standard error).

**Table 1 healthcare-11-01520-t001:** Comparison of total number of patients considered at admission, according to the risk of sarcopenia.

	Risk of Sarcopenia		*p*-Value
	Yes (Sarc-F ≥ 4) (*n* = 48)	No (Sarc-F < 4) (*n* = 87)	
Age, years	84.1 ± 6.7	80.2 ± 7.8	0.005
Female, *n* (%)	40 (83)	57 (65)	0.030
Body mass index, kg/m^2^	26.7 ± 4.9	26.9 ± 4.1	0.850
Type of fracture, *n* (%)			0.263
Intracapsular	18 (37)	39 (45)	
Subtrochanteric	9 (19)	8 (9)	
Pertrochanteric	21 (44)	40 (46)	
Osteosynthesis material, *n* (%)			0.905
Nail	30 (63)	52 (60)	
Partial prosthesis	16 (33)	30 (34)	
Total prosthesis	2 (4)	5 (6)	
Charlson index	5.3 ± 1.6	4.5 ± 1.3	0.003
Delirium	5 (10)	5 (6)	0.326
Number of physiotherapy sessions	6.3 ± 3.4	4.9 ± 3.1	0.017
Days of hospital stay	16.1 ± 8.7	12.9 ± 6.7	0.020
Patient location before, *n* (%)			0.721
Home	44 (92)	82 (94)	
Elderly residence	4 (8)	5 (6)	
Patient location after, *n* (%)			0.250
Home	36 (77)	76 (87)	
Social health centre	5 (10)	6 (7)	
Elderly residence	6 (13)	5 (6)	
Functional tests			
Katz			
Average ± s.d.	4.1 ± 1.4	4.8 ± 0.7	<0.001
Katz, *n* (%)			0.003
Severe disability (0–2)	7 (15)	2 (2)	
Moderate disability (3–4)	14 (29)	15 (17)	
Mild or no disability (5–6)	27 (56)	70 (81)	
Lawton			
Average ± s.d.	3.6 ± 2.5	5.5 ± 2.6	<0.001
Median (P_25_; P_75_)	4 (1; 5.8)	6.0 (3.2; 8)	<0.001
Lawton, *n* (%)			0.001
Severe dependence (0–2)	20 (42)	15 (17)	
Mild dependence (3–5)	16 (33)	24 (28)	
Independence (6–8)	12 (25)	48 (55)	
FAC			
Average ± s.d.	3.8 ± 0.9	4.5 ± 0.6	<0.001
FAC, *n* (%)			<0.001
Walking with aid (0–1)	2 (4)	-	
Supervised walking (2–3)	12 (25)	4 (5)	
Independent walking (4–5)	34 (71)	83 (95)	
Cognitive Impairment			
Pfeiffer			
Average ± s.d.	4.7 ± 2.4	3.5 ± 2.5	0.008
Median (P_25_; P_75_)	5 (3; 7)	3 (1; 5)	0.007
Pfeiffer, *n* (%)			0.098
No impairment (0–2)	11 (23)	34 (39)	
Mild impairment (3–4)	10 (21)	23 (26)	
Moderate impairment (5–7)	21 (44)	23 (26)	
Severe impairment (8–10)	6 (12)	7 (8)	

s.d. = standard deviation. P25; P75 = Percentile 25 and 75, respectively. *n* = number of cases. FAC = Functional Ambulation Classification. Katz index = basic daily life activity. Lawton and Brody scale = instrumental activity daily life. Pfeiffer = Quantitative cognitive function.

**Table 2 healthcare-11-01520-t002:** Information on the patients with follow-up during the study period (admission, hospital discharge, and one year) according to the Sarc-F score at admission.

				*p*-Value *		
	Admission	Discharge	One Year	Time (Linear; Quadratic)	Time * Sex	Time * Sarc-F
Katz, points				(0.077; <0.001)	0.118	0.008
Sarc-F < 4	4.7 ± 0.7	1.9 ± 0.9	4.1 ± 1.3			
Sarc-F (≥4)	3.9 ± 1.5	1.4 ± 0.8	2.6 ± 1.6			
Katz, *n* (%)				(0.001; <0.001)	0.282	0.084
Sarc-F < 4						
Severe disability	2 (3)	61 (79)	12 (16)			
Moderate disability	14 (18)	15 (20)	21 (27)			
Mild or no disability	61 (79)	1 (1)	44 (57)			
Sarc-F (≥4)						
Severe disability	7 (19)	31 (86)	19 (53)			
Moderate disability	10 (28)	5 (14)	11 (30)			
Mild or no disability	19 (53)	-	6 (17)			
Lawton, points				(0.032; <0.001)	0.001	0.012
Sarc-F < 4						
Average ± s.d.	5.4 ± 2.6	0.8 ± 0.5	3.7 ± 2.7			
Median (P_25_; P_75_)	6 (3.1; 8)	1 (1; 1)	4 (1; 6)			
Sarc-F (≥4)						
Average ± s.d.	3.4 ± 2.5	0.4 ± 0.5	1.7 ± 1.9			
Median (P_25_; P_75_)	3 (1; 5.8)	0 (0; 1)	1 (0; 2)			
Lawton, *n* (%)				(0.041; <0.001)	0.099	0.001
Sarc-F < 4						
Severe dependence	14 (18)	77 (100)	31 (40)			
Mild dependence	21 (27)	-	25 (33)			
Independence	42 (55)	-	21 (27)			
Sarc-F (≥4)						
Severe dependence	16 (44)	36 (100)	28 (78)			
Mild dependence	11 (31)	-	5 (14)			
Independence	9 (25)	-	3 (8)			
FAC, points				(0.394; 0.035)	0.001	0.183
Sarc-F < 4	4.5 ± 0.6	2.3 ± 0.6	4.3 ± 1.2			
Sarc-F (≥4)	3.7 ± 1.0	1.1 ± 0.4	3.0 ± 1.6			
FAC, *n* (%)				(0.002; 0.003)	0.022	0.215
Sarc-F < 4						
Walking with aid	-	5 (6)	5 (7)			
Supervised walking	4 (5)	72 (94)	11 (14)			
Independent walking	73 (95)	-	61 (79)			
Sarc-F (≥4)						
Walking with aid	2 (5)	31 (86)	8 (22)			
Supervised walking	10 (28)	5 (14)	9 (25)			
Independent walking	24 (67)	-	19 (53)			

* *p*-values obtained from the Repeated Measures model when comparing the functional tests given in points (continuous) and the Generalized Linear Model with multinomial distribution when it is categorical scale (percentage), introducing as factors sex, the coded Sarc-F, and the covariate age at admission. s.d. = standard deviation. P25; P75 = Percentile 25 and 75. FAC = Functional Ambulation Classification. Katz index = basic daily life activity. Lawton and Brody scale = instrumental activity daily life.

**Table 3 healthcare-11-01520-t003:** Study of the functional tests one year after performing the hip operation based on the information at admission.

	Coefficient	s.e.	t	*p*-Value
Katz				<0.001
Constant	4.436	0.238	18.633	<0.001
Sarc-F (≥4)	−1.392	0.283	−4.921	<0.001
Delirium	−1.014	0.543	−1.866	0.065
Pfeiffer (ref. no impairment)				
Mild impairment (3–4)	0.002	0.338	0.006	0.995
Moderate impairment (5–7)	−0.675	0.332	−2.031	0.045
Severe impairment (8–10)	−1.111	0.471	−2.357	0.020
Lawton				<0.001
Constant	13.718	2.063	6.649	<0.001
Sarc-F (≥4)	−1.237	0.426	−2.906	0.004
Delirium	−1.950	0.804	−3.670	<0.001
Age	−0.110	0.026	−4.204	<0.001
Pfeiffer (ref. no impairment)				
Mild impairment (3–4)	−1.216	0.503	−2.417	0.017
Moderate impairment (5–7)	−2.209	0.495	−4.463	<0.001
Severe impairment (8–10)	−2.332	0.720	−3.239	0.002
FAC				<0.001
Constant	4.715	0.249	18.940	<0.001
Sarc-F (≥4)	−1.224	0.263	−4.649	<0.001
Delirium	−1.342	0.504	−2.662	0.009
Sex (Male)	−0.843	0.275	−3.068	0.003
Pfeiffer (ref. no impairment)				
Mild impairment (3–4)	0.186	0.312	0.596	0.553
Moderate impairment (5–7)	−0.417	0.305	−1.371	0.173
Severe impairment (8–10)	−0.707	0.353	−2.160	0.036

s.e. = standard error. FAC = Functional Ambulation Classification. Katz index = basic daily life activity. Lawton and Brody scale = instrumental activity daily life. Pfeiffer = Quantitative cognitive function.

## Data Availability

The data presented in this study are available on request from the corresponding author. The data are not publicly available due to patient confidentiality.

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
