# Peer review of "Functional Status of Patients over 65 Years Old Intervened on for a Hip Fracture One Year after the Operation"

_healthcare, 2023, doi:10.3390/healthcare11101520_

Round 1

Reviewer 1 Report

This is a prospective observational study with 135 patients over 65 years of age aimed at early evaluation of the functional status one year after hip fracture surgery and the influence of sarcopenia. The study is interesting. There are, however, several areas in the manuscript that deserve improvements. Please find them below.

1.       Mentioning the statistical tests in the abstract is not a common practice.

2.       Spell out the abbreviation “FAC” the first time it has been mentioned in the abstract.

3.       I am curious why an assessment of the cognitive functions was considered one year after hip fracture surgery. This should be made clear.

4.       43% moderate-severe cognitive impairment. You may need to spare more space to interpret This finding in the discussion. Participants were over 65 years old? Should hip surgery affect their cognitive functions one year later or the changes might have been related to aging?

5.       You may need to refer to the study population (i.e., patients over 65 years old who underwent hip fracture surgery) in the title.

6.       Functional assessment after a year of hip fracture surgery is a rather late follow-up, to what the author claimed. The objective and conclusion statements should be adjusted. The term “early” is likely inappropriate.

7.       Although a sample size of 135 participants seems reasonable, I am not sure if the study is powered enough for the study design and outcome measures.

8.       Please, provide a further description of all measuring instruments. The domains of function measured by each. How some apply differently to male and female participants (e.g., Lawton & Brody Scale).

9.       Should assessing the status telephonically after one year be an effective follow-up? I understand that this may reduce the loss to follow-up of participants, however, authors should consider more practical methods in their future work.

10.   The statement of the study objective is not consistent throughout the introduction and discussion. Revise, please.

11.   Results are explicitly presented. The tables and figures are informative. I was able to follow the results based on the study objectives.

12.   The discussion reads well. However, it would benefit from focusing on the relevance to practice and offering the authors’ perspectives on how intended professionals make use of these findings.

13.   References No 22, 23, 24, and 41 are fairly outdated. Could you provide credit to more recent citations?

The English Language is good. Additional editing might correct some little flaws and enhance the sentence structure.

Reviewer 2 Report

I am very pleased to have the opportunity to review this manuscript. This manuscript is titled "Functional status of patients intervened on for hip fracture one year after the operation" and is about the functional recovery after hip surgery. It is an important report that mentions the influence of sarcopenia, which has been attracting attention in recent years, and I think it is academically significant.

I have read the manuscript and would like to comment as follows.

1.(Introduction)

What is the contribution of correlating functional status one year after surgery? For prognostic purposes, longer-term observation than 1 year may be necessary. It may also be necessary to establish a control group. Please explain the rationale for the one-year observation period and the new findings from this study.

2.(2.1. Procedure)

At the time of the one-year survey, do we have information on how each patient was doing during that period of time? In other words, whether or not they are receiving any services, including physical therapy, after discharge from the hospital may have an impact on their condition at one year.

3.(Materials and Methods)

When mentioning sarcopenia, it may be necessary to consider nutritional status. If it has been considered, it should be stated as such. If not, why not describe it as a limitation of the study?

4.(Results)

It was found that the characteristics of each evaluation index differed between the sarcopenia and non-sarcopenia groups, which were classified based on their condition at the time of admission. Furthermore, differences between the sarcoma and non-sarcoma groups have also been identified in the functional assessment after one year. Is the fact that the scores are different at the time of the initial evaluation taken into account in the comparison? I could not determine the above from the text. For example, how about examining by rate of change or correcting for relevant factors as covariates?

Reviewer 3 Report

Comments to 2388198

The manuscript by Pablo A. Marrero-Morales et al studied  the functional status of  elderly patients one year after hip fracture operation is well designed and written by using a perspective observation approach. The results established several risk factors related with functional recovery including Sarcopenia risk, daily activities, and cognitive impairment. Data presentations are clear. The established model of prediction is meaningful and warrants clinical application for patients with hip fractures.  

Only few minor types need to be corrected as below:

Page 6 line 227, “O” should be “or” ?.

Page 10, line 339: “being more evident in the one that presents it”. Not Grammarly smooth.  May be “ being more evident in the patients with sarcopenia”.

Page 11: Line 34,  first word “within”  should be deleted.

English overall is good. 

Round 2

Reviewer 2 Report

I checked the reply and confirmed that it was corrected appropriately.